# Systematic Review of the Health and Equity Impacts of Remediation and Redevelopment of Contaminated Sites [note 1]

**DOI:** 10.3390/ijerph19095278

**Published:** 2022-04-26

**Authors:** Danielle Sinnett, Isabelle Bray, Gergő Baranyi, Matthias Braubach, Sinaia Netanyanhu

**Affiliations:** 1Centre for Sustainable Planning and Environments & WHO Collaborating Centre for Healthy Urban Environments, University of the West of England, Bristol BS16 1QY, UK; 2Centre for Public Health and Wellbeing & WHO Collaborating Centre for Healthy Urban Environments, University of the West of England, Bristol BS16 1QY, UK; issy.bray@uwe.ac.uk; 3Centre for Research on Environment, Society and Health (CRESH), Institute of Geography, The University of Edinburgh, Edinburgh EH8 9XP, UK; gergo.baranyi@ed.ac.uk; 4European Centre for Environment and Health, World Health Organization Regional Office for Europe, Platz der Vereinten Nationen 1, 53113 Bonn, Germany; braubachm@who.int (M.B.); netanyahus@who.int (S.N.)

**Keywords:** brownfield, regeneration, human health risk assessment, contaminated land, urban soil

## Abstract

(1) Background: Globally there is a vast legacy of contaminated sites from past industrial, commercial and military activity, waste disposal, and mineral extraction. This review examined the extent to which the remediation of contaminated sites reduces health risks to new and existing populations. (2) Methods: Standard academic databases were searched for papers that reported on health-related outcomes in humans following remediation and redevelopment of contaminated sites. Title/abstract screening, followed by full-text screening identified sixteen papers that met the eligibility criteria. (3) Results: Most studies were set in the United States of America and reported changes in blood lead concentrations in children, following soil remediation and, in some cases, public health campaigns to reduce exposure. Two further studies examined the impacts of remediation on soil contaminated with chromium and sediments contaminated with polychlorinated biphenyls (PCBs). (4) Conclusions: Overall, the evidence suggests that remediation via removal, capping, and replacing soil, and planting vegetation is effective at reducing concentrations of lead and chromium in blood and urine in children. There is also evidence that sediment dredging can reduce PCB concentrations in umbilical cords in infants. Study designs are relatively weak and some recommendations are provided for those wishing to examine the health impacts of remediation.

## 1. Introduction

Land can be affected by contamination from current or previous land use, including on site or nearby land uses, such as from chemical, textile, timber, printing and coating industries, generation of energy, management of waste, mining and processing of metals, transport and engineering [1]. These industries may result in contamination of land through disposal of waste materials, accidental spillage or release of pollutants or deposits from air pollution [2,3]. Many countries have introduced regulations to reduce the risk of land becoming contaminated, but activities prior to regulation have left a global legacy of contaminated sites. The European Environment Agency (EEA) estimated that there were 2.8 million potentially contaminated sites across the EU-28 in 2018, and that only approximately 650,000 of these had been formally registered, representing an increase of 76,000 since 2014 [4]. Due to the vast health risks represented by contaminated sites, the World Health Organization (WHO) Regional Office for Europe has started work to assess the health risks of contaminated sites [5]. Based on the available evidence, waste and contaminated sites were established as one of the priority topics for the Sixth Ministerial Conference on Environment and Health (Ostrava, Czechia, June 2017), where member states committed to tackling the environment and health impacts of waste management and contaminated sites [6]. Following this commitment, the WHO Regional Office has developed a planning brief on protecting health through urban redevelopment of contaminated sites to support national and local governments in related efforts [7]. Furthermore, in November 2021, the European Commission adopted a new European Union (EU) soil strategy, aiming at concrete measures to protect and restore soils, and announced the establishment of a new Soil Health Law by 2023 [8].

To prevent environmental threats or health risks from contaminated sites, individual countries have developed legal definitions and risk assessment frameworks for the management of contaminated land [4,9]. Contaminated sites are defined here as: “areas having hosted or being affected by human activities which have produced environmental contamination of soil, sediment, surface or groundwater, air, or food-chain, resulting or being able to result in harm to human health, the environment or ecological systems” (adapted, based on Martuzzi, Pasetto and Martin-Olmedo [10]).

Contaminants include metals (e.g., cadmium, lead), metalloids (e.g., arsenic, antimony), organic substances (e.g., oils, polyaromatic hydrocarbons (PAHs), benzene, toluene, ethylbenzene and xylenes (BTEX), methyl tert-butyl ether (MTBE), polychlorinated biphenyls (PCBs), pesticides, chlorinated solvents, dioxins, volatile and semi-volatile organic compounds (VOCs; S-VOCs), and tars), acids and alkalis, asbestos, gases (e.g., methane), and radioactive substances. The health impacts of contaminants include decreased life expectancy, cognitive impairment and neurological damage, cancers, adverse impacts on respiratory, renal, reproductive and digestive systems, reduced foetal growth and miscarriages, and acute poisoning [1,2,11,12,13,14,15,16]. It is difficult to assess the impacts of contaminated land on human health, due to confounding by individual and area-level factors, such as other sources of pollution (e.g., air pollution), area deprivation, socioeconomic status and behaviours [17]. Furthermore, they usually affect whole local populations making it difficult to identify appropriate control groups.

Land affected by contamination is prioritised for remediation when it is adversely impacting, or is likely to adversely impact, a ‘receptor’. Receptors may include humans, water bodies, crops, property, and specific ecological systems. Contaminated sites do not necessarily pose a risk to receptors, for example, if they are not close by, or the contaminants are sufficiently immobile in the soil [13,16,18]. While contaminated land located in, or close to, urban centres may present a risk to people living nearby, the multiple sources of contamination in such areas often make it difficult to demonstrate an association between a source and a health impact, even if a risk assessment has identified that there is likely to be such an impact [17,19]. Closed sites are also often unattractive, derelict and a waste of land (i.e., brownfields), and their redevelopment can help prevent urban sprawl [20]. It is often due to redevelopment that contaminated sites are remediated [21]. For example, in England and Wales around 90% of sites remediated are through redevelopment [22]. A risk assessment will consider not only the risks presented by the site in its existing state, but also any risks that may occur through redevelopment (e.g., to site workers or new populations moving on to the site), and remediate taking into account newly introduced pathways and receptors. It is important to acknowledge that risk assessment of contaminated sites is often based on whether soil, water and/or indoor air concentrations are above reference values set to protect human or environmental health [9,13,18,19,23]. Where concentrations are found to be above these levels then a remediation strategy is developed [23], or further risk assessment takes place, that may involve additional site investigation, modelling or measurement of health-related outcomes (e.g., biomonitoring). A human health risk assessment (HHRA) often includes an exposure assessment to estimate the relative importance of different pathways to overall exposure [9].

Where a population is identified as being at risk, remediation may be combined with a public health campaign to reduce exposure (e.g., by reducing household dust exposure, or by cessation of vegetable growing) [15], or residents may be advised to relocate [24]. In extreme cases, such as Love Canal, United States of America (USA), the residents may be permanently evacuated, homes demolished and the contamination managed [11]. Remediation technologies can be described as physical (e.g., placing a barrier between a source and a receptor), chemical or biological methods (e.g., removing, degrading or immobilizing the contaminants), for example, using chemical additions to soil [1].

Remediation strategies normally include post-remediation monitoring to ensure that the objectives have been achieved [25], but these are usually focused on environmental media (e.g., soil, water or dust) and do not commonly include monitoring of human health outcomes. Given that most sites are remediated as part of the redevelopment process there is not a ‘before’ population on which to examine pre-remediation health outcomes, although neighbouring populations may be included in monitoring programmes.

Monitoring reports are often held by consultants and not in the public domain [26]. Indeed, several recent reviews have reported that the lack of full-scale remediation studies in the literature is hampering an evaluation of the outcomes of remediation [19,27,28]. Instead, these reviews focused on laboratory experiments or pilot demonstrations on site, which tend to dominate the literature due to the focus on development of new remediation technologies. Those publications that examine outcomes related to human health tend to cover the pre-remediation risk assessment and/or rely on data modelling exposure pathways or risk (e.g., [2,3,20,23]). However, there is a need to examine the evidence related to full-scale remediation and redevelopment of contaminated sites and their efficacy in terms of reducing harm to human health.

The aim of this systematic review is, therefore, to provide a robust assessment of the evidence for the effectiveness of remediation and redevelopment of contaminated sites in reducing environmental and health risks. It focuses on the redevelopment of contaminated sites to new residential neighbourhoods and public or recreational functions. Due to the small number of studies that evaluate post-redevelopment outcomes with specific reference to the remediation of contamination, the scope includes the remediation of contaminated sites to reduce or prevent health and environmental risks in existing urban areas in close proximity to the site. The review sought to answer the question: to what extent does remediation, and any subsequent redevelopment, of contaminated sites reduce environmental and health risks to new and existing populations and ecological systems, and are there any effects on equity in terms of the distribution of risks and outcomes?

## 2. Methods

This systematic review focuses on contaminated sites of any kind. The study followed a systematic search strategy, followed by title and abstract screening, full-text screening, data extraction and quality appraisal, following the Preferred Reporting Items for Systematic Review and Meta-Analysis (PRISMA) guidelines [29]. Since the review was carried out in a short space of time, it was not possible to develop and register a review protocol in advance, as suggested by guidelines for systematic reviews [30].

### 2.1. Search Strategy

The search strategy (Table 1) combined three sets of terms relating to the remediation and development of contaminated sites, based on De Sario et al. [31]. The first set of terms focussed on the contaminants (e.g., lead, polyaromatic hydrocarbons) and the land uses leading to contamination (e.g., mining, industrial). The second set of terms corresponded to remediation and redevelopment of these sites. The third described health-related outcomes (e.g., mortality, cancer). Title, abstract and keyword searches were carried out in Ovid (Embase, Medline, Global Health, PsycINFO, Cab Abstracts), Scopus, Open Grey, and ProQuest (theses database, ASSIA) on 4th June 2020. We used Scopus instead of Web of Science due to institutional subscription, because it is more international and interdisciplinary, both of which are important for our review, has better coverage of biomedical research [32] and includes health sciences [33]. We chose to include Open Grey as we expected that some evaluations relevant to our review might not have been published.

### 2.2. Eligibility Criteria

A total of 6903 papers were selected for title and abstract screening based on the following inclusion criteria: the site was known to have soil contamination as a result of a land use or disposal of waste on or adjacent to the site, or deposition of pollution from nearby land uses AND remediation of some or all of the site (by any means) had taken place AND there was an evaluation of health-related outcomes for new or existing populations following remediation and/or redevelopment.

Studies. were excluded if: they reported on exposure from general sources (e.g., vehicular traffic) OR they only reported changes following laboratory or field experiments, rather than full scale remediation OR they reported on the outcomes of modelling and/or risk assessments where risk to humans was estimated either before or after remediation had taken place OR they focussed on post-development outcomes relating to regeneration (e.g., employment, inward investment, deprivation), rather than those associated with the remediation of contamination.

### 2.3. Data Extraction

Screening was carried out by DS, and a 15% sample was screened by a second reviewer (IB). Few papers met the inclusion criteria following title/abstract screening (*n* = 50; 14 included new development, 36 considered existing populations). Following title/abstract screening each paper was full-text screened by DS against the above criteria. Reference lists were also searched for additional studies (see flow diagram in Figure 1).

Sixteen papers were included in the systematic review. Data were extracted from each paper using a standard proforma: Contaminant(s) of interest and source(s) of contamination; Remediation technique(s); Objective(s); Location; Study design and comparator; Sample size and population(s); Considerations of equity; Timing of sampling; Approach, methods, design; Outcome(s); Results (including confidence interval and *p*-values); Limitations/risks/bias. Data extraction was carried out independently by DS and IB and evidence synthesis was carried out by DS.

### 2.4. Quality Appraisal

Quality assessment was carried out using the Effective Public Healthcare Panacea Project’s Quality Assessment Tool for Quantitative Studies [34]). This tool assesses the quality of studies against a series of criteria, including study design, risk of bias, recruitment and withdrawal of participants (a detailed method can be found in the link above). Quality assessment was done independently by IB and DS and there was strong agreement between the assessments (94% for overall scores and 82% for individual components). Disagreement between scores related only to the representativeness of the sample study design (where not stated explicitly). All studies were judged as ‘weak’ using this tool (see Appendix A) and therefore studies were not excluded based on quality. 

The studies fell into three categories, used to structure the results of the review:

Studies that examine health-related outcomes for new residents following remediation and redevelopment of contaminated land, all of which focussed on lead (*n* = 3) [35,36,37];

Studies that examine health-related outcomes for children in existing neighbourhoods resulting from exposure to lead (*n* = 8) [38,39,40,41,42,43,44,45] and chromium (*n* = 1) [46] following remediation and public health campaigns;

Studies that examine health-related outcomes in existing populations following remediation of contaminated land (*n* = 4) [47,48,49,50].

## 3. Results

### 3.1. Study Characteristics

Sixteen studies were included in the evidence synthesis (Table 2). The majority were based in the USA (*n* = 9). The remaining studies came from Australia (*n* = 1), Canada (*n* = 2), Nigeria (*n* = 1), Chile (*n* = 1), Italy (*n* = 1) and Finland (*n* = 1). Most of the studies (*n* = 12) reported on remediation of sites contaminated with lead originating from smelters in Finland [35], the USA [38,39,40,47], and Canada [41], from a lead reclamation plant in Canada [42], from informal gold mining in Nigeria [43], a copper mine in the USA [36], a lead mine in Australia [44] and from several sources in the USA [37,45]. Another study reported on the remediation of chromium waste sites [46] and another on the dredging of a harbour to reduce exposure to polychlorinated biphenyls (PCBs) [48], both in the USA. The remaining studies reported on sites contaminated with multiple metals. One reported on blood lead levels (BLLs) from a waste disposal site in Chile [49] and the other on cadmium, chromium, copper, manganese, lead and zinc from mining and industrial sources in Sardinia, Italy [50].

Most studies were cross-sectional (*n* = 8) [35,36,39,43,46,47,49,50]. Others included a randomised control trial (*n* = 1) [45], cohort study (*n* = 2) [37,48], pre-post-remediation (*n* = 1) [42], case-control (*n* = 1) [38] and interrupted time series (*n* = 3) [40,41,44].

The majority (*n* = 15) studied children living near sources of contamination, reflecting the vulnerability of this group to exposure from soils due to hand-to-mouth behaviours and the toxicity of contaminants such as lead [51]. One study examined BLLs in adults [50].

Most studies (*n* = 14) reported the results of human biomonitoring (HBM) of contaminants in blood [35,36,37,38,39,40,41,42,43,44,45,47,49,50]. Two studies reported concentrations in urine [46,49] and one in umbilical cord serum [48]. Concentrations were compared with a control group, post-remediation concentrations, or thresholds set to protect health (e.g., a BLL of 5 µg/dL) [35,37,38,39,40,41,42,43,44,45,46,47,49,50]. Some studies used the proportion of children exceeding these concentrations as a population-level outcome to assess the impact of remediation [36,37,39,40,42,47]. Remediation usually aimed to reduce soil concentrations to an acceptable level, but in one study this was reported as an additional outcome [43]. Household dust is an important exposure pathway from contaminated soil to humans, and two studies reported concentrations in household and/or day care centre dust as an outcome [35,47]. One study examined the impact of contaminant exposure and subsequent remediation on cognitive performance in children born pre- and post-remediation of a waste disposal site [49].

The evidence synthesis is based on the broad categories outlined above. The studies varied in the level of detail provided on the remediation and/or redevelopment of the sites and this information is included in the evidence tables where available.

### 3.2. Remediation Followed by Redevelopment

Three studies examined BLLs in children following remediation and redevelopment. The focus of these studies was the impact of the remediation, so the description of the redevelopment is less detailed (Table 3). All three studies report declining BLLs following remediation [35,36,37]. In adults, lead can damage several organs and is associated with hypertension, diabetes and heart disease, but it is the impact on children that is most commonly cause for concern-it is associated with reduced cognitive performance and increased behavioural problems [37].

Remediation generally involved the removal and replacement of surface soils although the criteria used to decide which soils should be remediated and the reasons for doing so varied between the studies. Although they all include redevelopment in some form, the focus of these studies is BLLs in existing populations.

The objectives of the studies also differ. The study in Finland examined the different exposure routes from remaining contaminated soils to ascertain whether children living in the area were at risk and to inform further remediation [35]. The study in New Orleans investigated persistent disparities in BLLs to identify areas where remediation was still required post-Katrina [37]. The study in Butte assessed changes in BLLs following a remediation programme [36].

The study in Finland [35] collected primary data on BLLs for the study. Conversely the two USA studies used data from existing population screening programmes [36,37].

Although these three studies report on changes in BLLs following redevelopment as well as remediation, their findings are primarily concerned with the impact on existing populations living on, or near to, contaminated land. They provide consistent evidence that soil remediation can lower BLLs of children living near contaminated sites. However, except for the study in Butte [36], the reporting of these studies makes it difficult to directly relate remediation to the changes in BLLs.

### 3.3. Existing Populations with Remediation and Public Health Intervention

There are nine studies that examine the impacts of remediation on existing populations without considering redevelopment of the site. In these situations, remediation is often carried out alongside a public health campaign with the aim of reducing exposure via household dust, which has a high proportion of soil, and reducing hand-to-mouth behaviours in young children.

These studies also tend to focus on lead contamination (Table 4). There is one study on chromium contamination in the USA [46], and five that consider lead exposure in the USA [45], Canada [41,42], Australia [44] and Nigeria [43]. The remaining three studies relate to lead contamination at the same site, Bunker Hill Superfund Site in Idaho, USA. These present the initial period of remediation [38], a detailed analysis of the impact of the different interventions [40], and a review of the remediation and its impact on BLLs [39].

In all nine studies the contamination came from a combination of waste disposal and industrial deposition. Again, the remediation methods mainly involved removal and capping of contaminated soil. The purpose of the studies was to evaluate the effectiveness of the remediation and public health campaigns, with some studies also assessing the impact of exposure pathways on levels of lead in blood, or chromium in urine, of local children [38,46].

Most of the studies presented here analysed BLLs collected as part of a screening programme. There are, however, two exceptions. The cross-sectional study in New Jersey measured chromium in household dusts and urine of 41 children from Lafayette Gardens (a public housing project surrounded on three sides by chromium disposal sites), and 23 children, matched by age and sex, from three comparator neighbourhoods, which included a public housing project and a more affluent neighbourhood [46]. The other exception was the study in Boston [45], which consisted of a randomised trial with three groups (groups 2 and 3 were later combined): those receiving soil remediation, home cleaning and interior loose-paint stabilisation (group 1; *n* = 54), those with only home cleaning and paint stabilisation (group 2; *n* = 51) and those receiving paint stabilisation (group 3; *n* = 47) [45].

Taken together, these studies provide consistent evidence that remediation of contaminated soils is effective at reducing both direct and indirect exposure to pollution in adjacent populations. The studies from Bunker Hill suggest that remediation of yards alone is not sufficient (the resuspension of soils from other locations leads to recontamination and inward migration means that new families moving into unremediated homes are at risk [39,40]). In addition, contaminated areas accessible by children lead to another direct exposure pathway [43,46]. One study in Bunker Hill [40] suggested that area-wide remediation was responsible for around three times more reduction in BLLs than individual yard remediation. These studies also demonstrate that, where populations are exposed to contamination and remediation programmes, public health campaigns are also effective at reducing exposure pathways. These campaigns provide information about the risks from contamination and the importance of cleaning dusts from homes, good personal hygiene, and discouraging hand-to-mouth behaviour and pica (eating dirt) in children [39,43]. Most of the studies do not report separately about the impact of public health campaigns, but the reduction in hand-to-mouth behaviour and pica [42], and the estimated effectiveness of the intervention [40], suggests that this can be an important strategy to reduce exposure while remediation is carried out.

### 3.4. Existing Population with Remediation Only

The remaining studies assessed the impact of remediation of contaminated soils on the health of existing populations. These studies reported on remediation only; there was no public health campaign reported, although this may have taken place (Table 5).

These studies relate to sites contaminated by waste disposal [48,49], mining and deposition of industrial metal pollution [47,50] and consider a mixture of inorganic pollutants [47,49,50] and polychlorinated biphenyls (PCBs) [48].

The remediation technologies described in these studies were heterogeneous. The abandoned waste disposal site in Arica, Chile was active between 1984 and 1999. During remediation the wastes were removed, the site capped and fenced off and the roofs of homes in the vicinity were cleaned [49]. In Midvale, Utah mine tailings were capped, and a remediation programme removed soil from yards where lead concentrations exceeded 500 mg/kg, replacing it with clean soil [47]. The PCBs in New Bedford Harbour Superfund Site, MA, USA were caused by industrial waste disposal, including a capacitor, between the 1940s and 1977; the remediation of this site involved dredging and removing contaminated sediments between 1994 and 1995 [48]. The study in the Sulcis-Iglesiente area of Sardinia, Italy focuses on metals from an active industrial area, including three mines restored in the 1990 s, two lead/zinc mines and one coal mine [50]. There is no detail on the exact remediation measures employed at these sites.

These studies suggest that remediation of soil contamination alone can result in reduced exposure and contribute to improvements in health outcomes. The studies in Midvale and Arica give more information on the remediation methods employed and report consistent decreases in the BLLs or improved cognitive performance [47,49]. The study in New Bedford Harbour suggests that, although dredging of contaminated sediments can be successful in helping to reduce PCB exposure, the volatility of these contaminants means that there may be an increase in exposure during remediation, which should be considered in the planning of remediation strategies [48].

## 4. Discussion

This systematic review found consistent evidence that remediation of contaminated soils is effective at reducing direct and indirect exposure to pollution in local populations. The evidence for health outcomes is weighted towards studies of lead contamination, although some studies consider chromium and PCB contamination. The studies from Bunker Hill [39,40] suggest that soil cover alone is not sufficient (there needs to be excavation of contaminated surface soils to prevent upward migration) and that area-wide remediation is required to prevent recontamination. The experiences at Bunker Hill also suggest that this is particularly important where existing populations are present and mobile, as families may move into contaminated homes, and that ongoing monitoring is essential so that further requirements for remediation activities can be addressed during the programme [34,35]. The Bunker Hill studies also demonstrate the importance of considering the sustainability of remediation, especially with climate change, as remediated material was exposed following extreme weather events [39,40].

Although it appears that soil remediation is largely responsible for the decline in soil, dust and blood concentrations of contaminants, there is also good evidence that public health campaigns are effective at reducing exposure pathways for existing populations [39,40,42]. Several studies have highlighted the importance of multiple public agencies collaborating in site investigation and remediation, and the development of public health campaigns [33,42,52]. The importance of communication with residents was also highlighted in terms of gaining their trust and maximising participation [39,53]. Data collection, remediation and public health campaigns should be sensitive to their needs and experiences [18,52].

Due to the nature of research on remediation methods for contaminated sites, very few studies in the academic literature report on outcomes related to human health following full scale remediation. Searches of grey literature (i.e., OpenGrey and theses database) did not result in any additional studies. It does not follow that these technologies are unreliable, only that the monitoring and evaluation programmes to date have not often been reported on in the academic literature [26]. Our literature search found many studies reporting on laboratory, modelling or pilot studies, rather than full-scale remediation, and the lack of long-term evaluations following full-scale remediation is highlighted as a weakness in the literature in previous reviews focused on the efficacy of remediation for environmental outcomes [19,27,28]. In some countries, detailed remediation strategies, with well-defined success criteria and long-term monitoring programmes, require regulatory approval prior to the commencement of remediation, which is often not easily accessible.

### 4.1. Strengths and Limitations

Although the studies reported here generally find that HBM of contaminants decline following remediation, only one study reported on a clinical outcome [49]. HBM is an effective method to assess exposure, as it is relatively simple to measure concentrations of contaminants, (e.g., of metals in blood or urine), and it may be able to detect elevated concentrations before the development of clinical outcomes [54]. Frameworks and threshold concentrations for soil, water and blood concentrations vary by country, which hampers comparison between studies. In this review, most of the studies were based in the USA and focused on lead contamination.

The studies included are concerned, in the main, with lead pollution, often from a single source. There is a lack of heterogeneity, both in terms of the contaminants and the remediation employed. This could be because of the complexities associated with measuring organic contaminants in biological matrices, along with their prevalence in the environment from multiple sources. There is a notable gap in the evidence in relation to the mental health impacts of site remediation and redevelopment.

There is a lack of longer-term follow-ups from the remediation programmes reported in these studies. Given that recontamination of soils was reported after a relatively short time period [39] and previous studies have found an upward migration of metals 30 years after restoration [18], it is important for screening programmes to continue and report on longer term outcomes. Similarly, there is a lack of long-term trials on full-scale remediation in the published literature. Few studies examined health-related outcomes following both remediation and redevelopment. This is likely to be due to several factors; for example, the evaluation of full-scale remediation is not common, and the timescales involved between remediation and redevelopment and the different stakeholders involved make a long-term study extremely challenging. There are methodological challenges associated with health assessments; populations are often not present pre-remediation, or are also exposed to various contaminants in the urban environment reducing the ability to assess the impacts of a particular site, and finding an appropriate control group is challenging. This may explain the preponderance of studies in the USA and Australia where population densities are lower and urban areas are more likely to have grown around one industry (making the pathway between site and population simpler to study). Additionally, it is not generally desirable to remind a new population of the contamination history of their site [17], especially if remediation objectives have been met and there do not appear to be ongoing health impacts.

Few of the studies explicitly considered equity in their study design or outcomes. It is well documented in environmental justice literature that disadvantaged groups are more likely to live near contaminated sites [54]. Some studies did collect data on socioeconomic variables and include these as confounders in their analysis [36,38,45,47,48,49,50], others collected these variables but did not include them in the analysis [43] or simply discussed their findings in relation to socioeconomic status [37,39,40,46]. Only one study appeared to consider equity at the design stage, including control groups from neighbourhoods with both similar and more affluent socioeconomic profiles [46]. Often, the populations located near contaminated sites tend to be more disadvantaged. This is important because socioeconomic factors are likely to affect the exposure to soils either directly (e.g., length of residence, diet) or indirectly (e.g., income), or they may be related to the outcome (e.g., maternal education, income in the study examining cognitive performance).

All the studies examining health-related outcomes were scored as ‘weak’ in the quality assessment. For certain criteria, this was due to the nature of these studies. For example, few studies were blinded, but this is because it is difficult to blind participants to the existence of contaminated sites, which tends to be well-known in a neighbourhood. Similarly, the outcome assessor is likely to know whether participants live close to a contaminated site, especially if household dust samples or door to door surveys are being carried out. This could be avoided by asking participants to attend a test centre or clinic, although this may reduce participation. Such criteria are of less importance for the studies in this review as they tend to use objectively-measured outcomes (e.g., BLLs) that are unlikely to be affected by the participant being aware of the research questions or the assessor knowing the exposure status of the participant. Where blinding becomes more important is when data are collected on behaviours related to exposure such as outdoor play, hygiene and hand-to-mouth activity, or lifestyle factors (e.g., diet, smoking). For example, in the study of chromium concentrations in children’s urine, playing on the waste sites was not reported by the children, even though it had been observed by the researchers. The authors suggest this may be because the parents were present during the survey [46]. The earlier study at Bunker Hill reported possible problems with recall of activities over a 9-month period. Parents of children with higher BLLs may be more likely to remember behaviours related to higher levels of exposure [38].

For other criteria, where papers scored poorly, this could have been avoided through better reporting of the study design. For example, it was usually not possible to tell how representative participants were of the target population, or the participation rate of those invited to take part. Studies that report analysis of data from population screening programmes (e.g., [37,40,43,48]) are more likely to be representative of the population but several studies did not report this information. Studies could also have provided more information about sample demographics; for example, the age range of the children in the study from New Orleans is not clear [37]. Participation in most of the studies was voluntary and several studies acknowledge selection bias as a limitation [38,39,40,45]. In Bunker Hill and Quebec, some parents did not want their children to be tested as they had low BLLs in previous years [39,42], meaning that the bias is likely to be towards those with higher BLLs [40]. One study from Bunker Hill highlights a practical and ethical consideration with many of these studies; the screening and remediation efforts target those most at risk of adverse health effects by design [40]. This bias in sampling is justified on ethical grounds [39], particularly where the need to treat a population overrides other considerations, as in the case of lead poisoning resulting from gold mining in Nigeria [43]. It was reported that if screening of BLLs at the Bunker Hill Superfund Site was characterised as an academic study, this had a negative effect on participation rates [40].

The number and reasons for participants dropping out of studies was not consistently reported. Another criterium where better reporting could have improved the quality ratings was related to the data collection methods, where it was not clear whether data collection methods were valid and/or reliable. Some studies, however, did report this information; for example, the precise methods of blood sampling and analysis with reference to standard protocols, or the use of standard tools (e.g., for cognitive performance).

Several studies collected data on confounders, but they did not always take them into account in the analysis or justify why some confounders were included in the analysis and others were not. Again, this could have improved the reporting of the study. In the case of behavioural factors this is particularly important, as it means that the effects of the public health programme cannot be separated out from those of the remediation. However, the later studies at Bunker Hill did assess the independent impact of remediation and public health campaigns [40].

Shortcomings in the design of studies of contaminated sites are difficult to address. There are not often resources available to provide a control group or randomised sample, and this would divert funds from assessing outcomes in at risk populations or taking remediation action [40]. Consequently, most of the studies considered here were cross-sectional. Some studies analysed outcomes before and after remediation [37,42,48], and were scored as ‘moderate’ on this criterium in the quality assessment. The analysis of the Bunker Hill Superfund site included a case-control study [38], and another study used samples from participants screened in multiple years to analyze an interrupted time series [40,41,44]. Most studies had relatively large sample sizes, but in a few cases the sample size affects the usefulness of the study [35,50]. Although the study in Boston was a randomised controlled trial [45], the analysis compared groups based on treatments received, rather than randomised groups (i.e., not ‘intention-to-treat’), thereby losing the benefits of randomisation.

Finally, the focus of several studies included in the review was on health-related outcomes, and therefore there was a lack of detailed information about remediation activities, thus restricting our ability to assess the effectiveness of specific remediation strategies.

### 4.2. Policy Implications and Future Research

Contaminated sites have raised considerable concern in many countries worldwide and affect local urban planning as well as global sustainability policies.

The large volume of industrial activities in such sites, associated with waste production, increased use of hazardous materials, as well as residual contamination in derelict military sites result in severe challenges for future use of potentially contaminated sites [55]. This is especially important for growing cities which have a continuous demand for land and cannot afford to leave abandoned sites (often referred to as brownfields) unused and undeveloped. Within the EU, land recycling and urban densification (such as converting industrial sites into urban functions and related infrastructure) accounted for only 13% of new developments and associated land take, identifying the increasing demand for land as a viable challenge for future sustainable development [56]. Therefore, redeveloping contaminated sites for urban functions is necessary for land recycling in many European countries. Reflecting this urgency, the remediation and redevelopment of contaminated sites has already been the focus of several activities and multilateral projects lead by the European Commission.

On a global scale, the problem of soil contamination is reflected in the Sustainable Development Agenda, which considers sustainable consumption and production patterns in the Sustainable Development Goal (SDG) 12. To feed the world sustainably, producers need to grow more food which requires healthy soils, unaffected by negative environmental impacts and degradation. The Agenda covers, among others, hazardous waste and chemicals, as well as extraction of natural resources [57]. Soil-related aspects are also covered (for example in relation to land degradation, ecosystem services or soil resources) in various other SDGs. Promoting sustainable production and management of resources including soil is also the objective of the circular economy concept, which aims to mitigate waste and pollution by keeping material resources in use and supporting natural material regeneration. Changing from a linear economy (take, make, dispose) to a circular economy (renew, remake, share) is therefore expected to support the attainment of SDG 12 [58]. The EU Green Deal, aiming to achieve a climate-neutral economy by 2050, thus includes the circular economy concept [59] and provides the policy context for the new EU soil strategy [8].

Interventions to redevelop contaminated sites therefore have both local and global policy relevance, and contribute directly and indirectly to social, environmental and health objectives. Research on the practical implementation, as well as the formulation, of adequate policy frameworks on the remediation and redevelopment of contaminated sites will be essential to support public authorities effectively tackle the local challenges that are posed by these sites.

This review highlights a real lack of long-term evaluations examining the impact of full-scale remediation in the academic literature. However, these studies do take place, often as a condition of remediation permits, and there is a need for the results to be published in the public domain, to enable a sharing of best practice. Where studies do exist, they tend to relate only to the impact of remediation on existing populations and/or do not include the redevelopment of the site. There are challenges in conducting this type of study, due to the timescales involved between remediation and redevelopment and the different actors involved at each stage in the process, as well as methodological challenges, including the lack of a pre-remediation ‘before’ population. However, if feasible, using administrative data (e.g., health and social outcomes, environmental data) could offer the opportunity to examine the impacts of contaminated site remediation and redevelopment, particularly where the impact of the site covers a large area, or where neighbourhood regeneration is concerned. Making use of secondary data, where available, may be more achievable than new studies. In addition, regeneration projects have previously been evaluated using natural experimental study designs (e.g., [60]) and this approach could also be taken to examine the impact of contaminated site redevelopment on health outcomes.

## 5. Conclusions

This systematic review examined the extent to which the remediation of contaminated land reduces health risks to new and existing populations. The papers included in the evidence synthesis suggest that there is good evidence that remediation via removal, capping, and replacing soil, and planting of bare soils, can reduce concentrations of lead and chromium in blood and urine in children. There is more limited evidence (from one study), that remediation of soils can improve cognitive performance in children living near waste dumps. There is also some evidence (also from one study), that sediment dredging can reduce PCB concentrations in umbilical cords in infants. However, the removal of soil is not a sustainable option for dealing with contaminated land and the preference now is to use technologies that reduce the environmental sequelae of remediation. Many other studies (not included in this review) examine the health outcomes associated with contaminated sites before remediation, and an opportunity exists to follow up some of these sites after remediation, with greatest priority being given to those that examine organic contaminants or inorganic contaminants other than lead. Similarly, results from long-term epidemiological or surveillance studies should be published whenever possible, to add to the existing evidence base.

## Figures and Tables

**Figure 1 ijerph-19-05278-f001:**
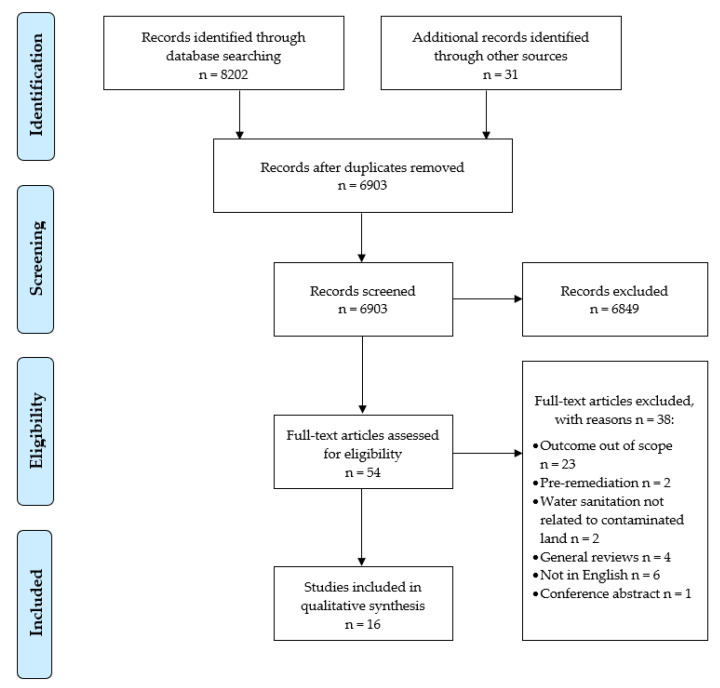
PRISMA flow diagram [25].

**Table 1 ijerph-19-05278-t001:** Terms in the search strategy.

#	Terms (adj = Adjacent)
1	((industr * OR mining OR mine OR quarries OR quarry OR waste OR incinerat * OR landfill * OR port OR harbor OR harbour OR ship OR dock OR superfund OR brownfield OR contaminat * OR site OR plant OR plants OR mill OR farm * OR agricult * OR land OR soil OR rail * OR derelict) AND (petro * OR pesticide * OR polymer * OR organochemical * OR colouring OR pharmaceutical OR paper OR metallurg * OR potter * OR fertilizer * OR footwear OR shoe * OR lindane OR plastic OR rubber OR detergent * OR lubricant * OR lubricating * OR weapon * OR glass OR iron OR steel OR asbestos OR fluoroedenite OR fluoro-edenite OR amosite OR erionite OR balangeroite OR tremolite OR crocidolite OR chrysotile OR serpentine OR antigorite OR anthophyllite OR actinolite OR ferroactinolite OR amphibole * OR lead OR cadmium OR arsenic OR nickel OR tin OR mercury OR chromium OR polyaromatic hydrocarbons OR cyanide OR polychlorinated biphenyls OR phenol OR BTEX OR benzene OR toluene OR ethylbenzene OR xylene OR trichloroethane OR vinyl chloride OR blue billy OR leblanc OR methane OR sewage sludge OR metal * OR gasworks OR filling station OR coal tar OR pulverised fly ash OR furnace bottom ash OR chemical OR oil OR chlorinate * OR volatile organic compound *))
2	“data mining”
3	1 NOT 2
4	(remediat * OR conver * OR renewal OR regenerat * OR rehabilitat * OR redevelop * OR reclamat * OR reuse OR re-use OR “clean-up *” OR restorat * OR cleanup * OR “clean * up *”)
5	3 AND 4
6	(health * OR mortality OR morbidity OR disease OR chronic OR infection OR syndrome * OR irritation OR ache * OR headache * OR nausea * OR sick OR pain OR sclerosis OR dent * OR neoplasm * OR tumor * OR tumour * OR cancer * OR lymphoma * OR leukaemia * OR leukemia * OR myelodysplas * OR myalgia * OR neuralgia * OR respirator * OR heart OR cardio * OR vascular OR stroke OR pulmonary OR lung OR respiratory OR renal OR kidney * OR bone OR digestive OR congenital OR reproductive OR semen OR retard * OR fetal OR foetal OR preterm OR pre-term OR miscarriage OR abort * OR pregnan * OR birth * OR death * OR neuro * OR muscl * OR urin * OR blood OR serum OR hair OR gland * OR throat OR eye * OR genotoxic * OR muta * OR biomonitoring OR bio-monitoring OR psych * OR brain OR skin OR epiderm * OR quality of life OR QoL OR satisfaction OR depression OR anxi * OR nervous OR stress OR sleep OR insomnia OR concentrat * OR cognitive)
7	5 AND 6
8	remove duplicates from 7

* Search terms are truncated by an asterisk. The term “industr*” will therefore find all words including this term, such as industry, industries, or industrial.

**Table 2 ijerph-19-05278-t002:** Studies meeting eligibility criteria included in the systematic review.

#	Author and Date	Title
35	Louekari et al., 2004	Reducing the risks of children living near the site of a former lead smeltery
36	Schoof et al., 2015	Assessment of blood lead level declines in an area of historical mining with a holistic remediation and abatement program.
37	Mielke et al., 2013	Environmental and health disparities in residential communities of New Orleans: The need for soil lead intervention to advance primary prevention
38	Maisonet et al., 1997	A case-control study to determine risk factors for elevated blood lead levels in children, Idaho
39	Sheldrake and Stifleman 2003	A case study of lead contamination cleanup effectiveness at Bunker Hill
40	Von Lindern et al., 2003	Assessing remedial effectiveness through the blood lead:soil/dust lead relationship at the Bunker Hill Superfund Site in the Silver Valley of Idaho
41	Hilts et al., 1998	Effect of interventions on children’s blood lead levels
42	Goulet et al., 1996	Results of a lead decontamination program
43	Tirima et al., 2016	Environmental remediation to address childhood lead poisoning epidemic due to artisanal gold mining in Zamfara, Nigeria
44	Boreland et al., 2008	Managing environmental lead in Broken Hill: a public health success
45	Aschengrau et al., 1997	Residential lead-based-paint hazard remediation and soil lead abatement: Their impact among children with mildly elevated blood lead levels
46	Freeman et al., 1995	The effect of remediation of chromium waste sites on chromium levels in urine of children living in the surrounding neighborhood
47	Lanphear et al., 2003	The effect of soil abatement on blood lead levels in children living near a former smelting and milling operation
48	Choi et al., 2006	Does living near a Superfund site contribute to higher polychlorinated biphenyl (PCB) exposure?
49	Burgos et al., 2017	Cognitive performance among cohorts of children exposed to a waste disposal site containing heavy metals in Chile
50	Madeddu et al., 2013	Blood biomonitoring of metals in subjects living near abandoned mining and active industrial areas

**Table 3 ijerph-19-05278-t003:** Studies reporting health-related outcomes for people living near remediated sites following remediation and redevelopment.

Study	Contaminant, Source and Remediation	Location	Study Design and Comparator	Population and Sample Size	Outcome Measures	Results
Louekari et al., 2004	Pb from a smelter (active 1929–1984). Surface soil replacement of area with Pb > 300 mg/kg prior to construction of new apartments, and in school and day care centre yards.	Tikkurila, Vantaa, Finland	Cross-sectional; comparators are unremediated areas and remote reference site. Secondary environmental data used to estimate exposure.	Population: 678 children aged 0–6 years living near smelter. Sample: 52 children from population (10 from unremediated area) and 11 from reference area.	Pb in air, water, lettuce, berries, and dust in household and day care centre. BLL	Air and water not important exposure route. Soil Pb ~20 mg/kg in remediated and reference areas, and >300 mg/kg in unremediated areas. blood lead levels (BLLs) in unremediated areas = 2.7 µg/dL, greater than in remediated areas (2.1 µg/dL) (*p* = 0.027); remediated areas comparable to reference areas of <2.0 µg/dL.
Mielke et al., 2013	Pb from multiple sources including industry and incinerator. Surface soil replacement of soils with >1000 mg/kg in homes; childcare centres (also had geotextile beneath soil), eleven public parks and 9 out of 10 public housing projects (during reconstruction).	New Orleans, LA, USA	Cohort analytic; comparator is low (< 100 mg/kg) soil Pb areas.	Pre-Katrina: 55,551 blood Pb samples from children. Post-Katrina: 7384 blood Pb samples from children.	BLL Percentage of children with BLL > 10 µg/dL. Percentage of children with BLL > 5 µg/dL.	Differences between soil and BLLs are significant (*p* < 0.001) between high Pb (median = 425 mg/kg) and low Pb (45 mg/kg) areas. BLLs reduced post-Katrina, in low (both medians = 3.0 µg/dL) and high Pb areas (3.0 vs. 5.6 µg/dL) (*p* < 0.001). Proportion of children with BLLs > 5 µg/dL reduced post-Katrina in low (7.5% vs. 24.8%) and high Pb areas (29.6% vs. 58.5%). Proportion of children with BLLs > 10 µg/dL reduced post-Katrina in low (3.0% vs. 1.0%) and high Pb areas (6.5% vs. 21.8%).
Schoof et al., 2015	Pb from copper mine. Stabilising, capping or removing waste and contaminated soils. Redevelopment to parks, activity centers and trails. Remediation ongoing.	Butte, MT, USA	Repeat cross-sectional; comparator is reference dataset.	2796 children aged 1–5 years (2003–2010) covering pre- and post-remediation.	BLL Percentage of children with BLL > 5 µg/dL and >10 µg/dL.	Decline in children with BLLs > 10 µg/dL from 3.4% to 1.5%; BLLs > 5 µg/dL from 33.6% to 9.5%. Butte BLLs greater than reference BLLs for 2003–2004 (mean = 3.48 vs. 2.05 µg/dL; *p* < 0.05), 2005–2006 (2.65 vs. 1.80 µg/dL; *p* < 0.05), and 2007–2008 (2.2 vs. 1.72 µg/dL; *p* < 0.05), but comparable for 2009–2010 (1.53 vs. 1.51 µg/dL). Butte BLL declined by 24% per 2-year increment, reference by 9% (*p* < 0.001). BLL greater in the uptown/historic area closer to mine, than ‘the flats’ area but only significant in 2007–2008 (*p* = 0.001) and 2009–2010 (*p* = 0.02).

**Table 4 ijerph-19-05278-t004:** Studies reporting health-related outcomes for people living near contaminated sites after remediation and public health campaigns.

Study	Contaminant, Source and Remediation	Location	Study Design and Comparator	Population and Sample Size	Outcome Measures	Results
Freeman et al., 1995	Cr, wastes from Cr manufacturing and refining. Sites capped and/or soil replaced (early 1990–late 1991). Public health campaign to reduce dust exposure.	Hudson County, NJ, USA	Cross-sectional; comparator is control areas (public housing and more affluent neighbourhoods) and pre-remediation data.	64 children: 41 children from Lafayette Gardens (public housing project surrounded on three sides by Cr waste sites) and 23 children from three control areas.	Cr levels in urine.	Cr concentrations in urine in children from Lafayette Gardens in Summer greater than controls (median 0.28 µg/L vs. 0.17 µg/L; *p* = 0.055) N.B. not significant in Winter. Cr concentrations in urine were age-dependent and related to home location. Controlling for personal rate of excretion and age, exposure status predicted Cr (regression coefficient = −0.347, SE 0.155, *p* = 0.03). Direct relationship between activities and Cr levels not confirmed.
Aschengrau et al., 1997	Pb from unspecified sources in soil and house paint. Phase 1: soil removal, addition of geotextile and soil replacement, dust abatement in homes and loose-paint stabilisation. Phase 2: soil remediation and interior and exterior paint remediation. Public health campaign.	Boston, MA, USA	Randomised control trial with three groups (Phase 1: 1: all treatments, 2: dust abatement and paint stabilisation, 3: paint stablisation; Phase 2: groups 2 and 3 offered soil remediation; all groups offered paint remediation.	152 children aged <4 years, with BLLs 7–24 µg/dL. Group 1 *n* = 54; group 2 *n* = 51; group 3 *n* = 47.	BLL	After Phase 2: group 1 children whose homes received only paint hazard remediation had mean blood lead levels (BLLs) 2.6 µg/dL (Confidence Interval (CI) = −0.6–5.9 µg/dL) greater than children who received no intervention. Group 2/3 children whose homes received paint hazard remediation and soil remediation had mean BLLs 1.4 µg/dL (CI = −0.73.5 µg/dL) greater than those whose homes had only soil abatement. After adjustment for confounders: group 1 children receiving paint hazard remediation had 6.5 µg/dL greater BLLs than those who did not (*p* = 0.05), there was no significant difference between the BLLs in group 2/3 children who did or did not receive paint hazard remediation (*p* = 0.36) suggesting soil remediation is effective.
Goulet et al., 1996	Pb from Pb reclamation plant (closed 1989). Asphalting plant yard, removing dust from roads and sidewalks, soil replacement, professional home cleaning (1989–1990). Public health targeting families with young children.	St-Jean-sur-Richelieu, QC, Canada	Cohort, one group pre-post remediation.	Children who lived 200 m from plant in 1991. Sample of 101 children aged 6 months-10 years (79.2% of population), 75 children in 1989 and 1991 sample.	BLL Percentage of children with BLL > 15 µg/dL.	In children who had participated in both surveys BLLs reduced from 9.7 µg/dL (95% CI = 8.6–10.9) in 1989 to 5.0 µg/dL (CI = 4.5–5.6) in 1991 (*p* < 0.001); children 6 months-5 years reduced from 9.8 µg/dL (CI = 8.6–11.2) to 5.5 µg/dL (CI = 4.9–6.3) (*p* < 0.001). In 1991, no children had blood Pb > 15 µg/dL compared with 21.3% in 1989. Percentage of children engaged in pica reduced from 35.5% (1989) to 18.8% (1991) (*p* = 0.004); putting things in mouth reduced from 46.2% to 31.7% (*p* = 0.03).
Hilts et al., 1998	Pb from a Pb/Zn smelter. Dust abatement including capping soils, use of a dust suppressant and greening. Public health campaign to raise awareness including provision of cleaning materials.	Trail, QC, Canada	Interrupted time series, screening programme, comparator is preceding year’s BLL.	Children aged 6–72 months. Sample size declined from 169 in 1989 to 46 in 1996.	BLL	BLLs reduced by 0.6 µg/dL (≈5%) per year between 1989 and 1996. In case management children, decline in BLLs (2.3 to 4.0 µg/dL) in the year following the intervention was significant for those receiving the intervention in 1991 (*p* < 0.001), 1992 (*p* < 0.001) and 1994 (*p* = 0.001).
Boreland et al., 2008	Pb from Pb mine. Capping of soil material, greening of bare soil. Public health campaign to raise awareness.	Broken Hill, New South Wales, Australia	Interrupted time series, screening programme, comparator is preceding years’ BLL.	Children aged 1–4 years, participation declined from 72% in 1994 to 46% in 2007.	BLL Dust Pb levels.	Mean BLLs reduced from 16.3 µg/dL (in 1991) to 5.8 µg/dL in 2007. Mean BLLs in the highest risk zone reduced from 27.3 µg/dL in 1991 to 8.3 µg/dL in 2007. Dust concentrations were significantly greater in 1991–1994, compared with 1995–1999 (*p* < 0.05).
Tirima et al., 2016	Pb from informal gold mining. Soil removal from residential and communal areas and ponds; landfill disposal. Public health campaign to reduce exposure through safer mining practices.	Eight villages, northern Nigeria	Repeat cross-sectional screening programme to identify at children at risk, alongside phased remediation.	4399 children aged < 5 years.	BLL Soil Pb levels.	Mean BLL reduced from 149 µg/dL to 15 µg/L over four-year period. Phase 1 (2 villages) soil Pb levels reduced by 98% and 96% to 83 mg/kg and 179 mg/kg respectively; 74 children screened before and during remediation had mean BLL of 149 µg/dL and 230 screened after remediation had mean BLL 76 µg/dL. Phase 2 (5 villages) soil Pb between 300 mg/kg and 1343 mg/kg reduced by 77% and 93% respectively; 3326 children screened and BLLs drop from ~48 µg/dL to ~25 µg/dL. Phase 3 (1 village with industrial area) mean soil Pb concentrations reduced by 87% from 670 to 90 mg/kg; BLL reduced from 25 to 15 µg/dL.
Maisonet et al., 1997	Pb from mine and smelter (closed 1981). Yard remediation; new yards remediated each summer since 1989.	Bunker Hill Superfund Site, ID, USA.	Case-control study, comparator is age and sex-matched children with BLL <10 µg/dL.	Population: 295 children aged 1–9 years. Sample: 138 participants (69 matched pairs).	BLL	Logistic regression: yard remediation associated with blood Pb levels after adjustment for income and education (Odds Ratio = 0.28, CI = 0.08–0.92, *p* < 0.05); pets in and out of house, hours spent playing outdoors, smoking inside house, child washes hands before bed, child puts dirt in mouth all non-significant.
Sheldrake and Stifleman 2003	As above. Pb from mine and smelter (closed 1981) Soil capping in parks and schools (1986), soil removal in residential yards, commercial properties and rights of way, and indoor dust based on child BLLs, soil concentrations and risk (1989-). Program expanded (1994-) to clean up adjacent parcels of land and areas with soils with >1000 mg/kg. Public health campaign to raise awareness and reduce exposure. By 2001, 80% of homes exceeding 1000 mg/kg were remediated.		Repeat cross-sectional, comparator is pre-remediation.	Children aged 9 months-9 years in the area offered annual BLL screening; percentage of eligible in sample exceeded 50% each year.	BLL Percentage of children with BLL > 10 µg/dL.	Percentage of children with BLL > 10 µg/dL; reduced over 80% un 1983 to 57.1% in 1998 to 4.4% in 2001 in 1-year-olds, 2-year-olds 60.9% to 9.8%, 3-year-olds 62.1% to 2.5%, 4-year-olds 36.8% to 4.3%; all children (<9 years) 46% to 3%.
Von Lindern et al., 2003			Interrupted time series, comparator is preceding year’s BLLs	Children in the area which is home to 7000 people in 5 communities; 230 to 445 children aged 9 months-9 years tested each year between 1988 and 2001; estimated as 50% of children on school records. 4000 paired BLL and environmental samples.	BLL Percentage of children with BLL > 10 µg/dL. Percentage of children with BLL > 15 µg/dL.	Percentage of children with BLL > 15 µg/dL reduced from 15% to 1.2% between 1988 and 2001; percentage with >10 µg/dL reduced from 45% to 3.1% between 1988 and 2001. Average BLL significantly different (*p* < 0.05) compared with preceding year in 1989–1994, and 1998. BLLs reduced 50–60% with greatest decrease corresponding with initial home yard remediation. Proportion of children living with contaminated yards (>1000 mg/kg) decreased from 80% in 1988–1989, to 43% in 1990 and 25% in 1991, fluctuated between 18–29% between 1992 and 1996 despite remediation of additional 551 homes (inward migration), by 1999 only 4% had contaminated yards. BLLs in control, remediation and public health intervention group reduced by 0.4 µg/dL, 2.5 µg/dL and 4.8 µg/dL respectively (*p* < 0.001). Suggests remediation reduces typical 2-year old’s BLLs by 7.5 µg/dL between 1989 and 2001 (1.7 µg/dL from individual yard, 5.6 µg/dL from community and neighbourhood), and public health intervention results in an additional 3.9 µg/dL reduction.

**Table 5 ijerph-19-05278-t005:** Studies reporting health-related outcomes for people living near contaminated sites after remediation.

Study	Contaminant, Source and Remediation	Location	Study Design and Comparator	Population and Sample Size	Outcome Measures	Results
Lanphear et al., 2003	Pb and As from a mine and smelter. Tailings capped, soil removed from yards with Pb concentrations <500 mg/kg and replaced with clean soil.	Midvale, UT, USA	Repeat cross-sectional, comparator is yards without remediation.	Children aged 6–72 months in 1989 (*n* = 112) and 1998 (*n* = 215)	BLLs As and Pb in soil and dust.	1989: Greater levels of As and Pb soil and dust concentrations (*p* = 0.0001), interior and exterior paint Pb concentrations (*p* = 0.004 and *p* = 0.006 respectively), and = blood lead levels (BLLs) in children (5.6 µg/dL vs. 3.9 µg/dL; *p* = 0.0001) in homes eligible for soil remediation compared with those that were not; 11% of children in homes eligible for yard remediation had BLLs > 10 µg/dL, compared with 2.6% in the control group. 1998: no significant differences between the intervention and the control groups for BLLs (3.0 µg/dL vs. 2.6 µg/dL), dust As and Pb concentrations and soil As concentrations; soil Pb concentrations greater in control homes compared with yard remediation (95 vs. 54 mg/kg; *p* = 0.02); 1% of children in homes with yard remediation had BLLs > 10 µg/dL. After adjustment for potential confounders (age, mouthing behaviour, socioeconomic status, year) BLL declined by 2.3 µg/dL (CI = 1.8–2.9 µg/dL), BLLs intervention group reduced 42.8% faster than control group (*p* = 0.14), BLLs declined faster in children aged 6-36 months (2.5 µg/dL, CI = 1.8–3.5 µg/dL; *p* = 0.03) than those aged 36–72 months (2.0 µg/dL, CI = 1.3–3.0 µg/dL; *p* = 0.03).
Burgos et al., 2017	Pb (with As, Cd, Cu) in abandoned waste site (active 1984–1999) Waste removed, roofs of homes cleaned, decontamination of other areas of the city, site fenced and covered (1999–2003).	Arica, Chile	Cross-sectional, comparator is children born post-remediation.	Population: 735 children aged 6–15 years Sample: 180 children selected at random.	BLLs As levels in urine. Cognitive performance (Wechsler Intelligence Scale for Children).	BLLs 2 µg/dL in both cohorts (*p* = 0.059), no significant difference in As in urine (*p* = 0.369). Cognitive performance greater in post-remediation cohort (91.1 points) compared with pre-remediation (81.9 points). Processing Speed Index and Absence of Distractibility Index were the only components that were not different between cohorts, other components were statistically significant. After adjusting for age, sex, maternal IQ and paternal education, the estimated difference in total IQ between cohorts increases (pre-remediation is reference): during remediation β = 9.97; 95% CI 0.82 to 19.13; post remediation β = 16.14; 95% CI 1.53 to 30.74.
Choi et al., 2006	PCBs, waste disposal from local industry (1940s–1977) Dredging of contaminated sediments (1994–1995).	New Bedford Harbour Superfund Site, MA, USA	Cohort analytic study of infants, comparator is infants born pre-remediation.	Population: 788 mother-infant pairs where mother >18 years old Sample: 720 (69 excluded).	Umbilical cord PCB levels; total PCBs, light PCBs, heavy PCBs, 51 congeners and PCB-118.	Multivariate models: maternal age and birthplace were the strongest predictors of ΣPCB levels (*p* < 0.001). Maternal consumption of organ meat and local dairy products was associated with higher, and smoking and previous lactation with lower, ΣPCB levels (*p* < 0.05). Infants born later in the study had lower ΣPCB levels than infants born earlier in the study. There was a 17% change (−3 to 40%) in ΣPCB for infants born before/during dredging compared with those born after dredging (*p* < 0.10).
Madeddu et al., 2013	Cd, Cr, Cu, Mn, Pb and Zn. Two former metal mines, one former coal mine, one active industrial area (mines restored 1990s).	Sulcis-Iglesiente, Sardinia, Italy	Cross-sectional study, comparator is a control area with no industry or mining.	Sample: 265 healthy adults.	Cd, Cr, Cu, Mn, Pb and Zn blood levels (BLs).	Participants within 5 km of active industrial site (*n* = 29) have greater BLs than those in the control area: Cd (*p* = 0.05), Cu (*p* = 0.031), Mn (*p* = 0.05) and Pb (*p* < 0.001); within 2 km of coal mine (*n* = 48): Cd (*p* = 0.036), Mn (*p* = 0.005), Pb (*p* = 0.006) and Zn (*p* = 0.005); within 4 km of Pb/Zn mine (*n* = 129): Cd (*p* = 0.041), Mn (*p* = 0.037), Pb (*p* = 0.005) and Zn (*p* = 0.004); and within 3 km of Pb/Zn mine (*n* = 32): Mn (*p* = 0.022). Those within 3 km of Pb/Zn mine restored (*n* = 32) have greater BLs than control: Cu (*p* = 0.019) and Pb (*p* = 0.011). Cd, Pb and Zn positively correlated with age.

## Data Availability

The data extracted from the reviewed articles are included in Table 3, Table 4 and Table 5.

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
