# Peer review of "Systematic Review of the Health and Equity Impacts of Remediation and Redevelopment of Contaminated Sitesâ€"

_ijerph, 2022, doi:10.3390/ijerph19095278_

Round 1
Reviewer 1 Report
Accept
Author Response
We note that there are no comments. Many thanks for your positive recommendation.
Reviewer 2 Report
The authors have significantly improved the text of the article and that version may be publishedAuthor Response
We note that there are no comments. Many thanks for your positive recommendation.
Reviewer 3 Report
The review paper raises the important topic of the impacts of contaminated land on human health. It follows the PRISMA methodology and a research process is comprehensively described.
Please distinguish more in the introduction what is the aim of the research and what are the research questions.
Please elaborate further the research background, present the current state of the research field. I suggest expanding the review with more articles. Maybe change the search criteria - 56 references is not enough for a review article.
Author Response
The review paper raises the important topic of the impacts of contaminated land on human health. It follows the PRISMA methodology and a research process is comprehensively described.
Many thanks for this positive feedback.
Please distinguish more in the introduction what is the aim of the research and what are the research questions.
We have added an aim for the review in to Lines 125 to 127, the research questions are still set out in Lines 132 to 135..
Please elaborate further the research background, present the current state of the research field. I suggest expanding the review with more articles. Maybe change the search criteria - 56 references is not enough for a review article.
We have addressed a similar comment already, this is a systematic review, and as such follows a specific methodology and would repeat our previous response here:
We agree that our review was only able to include a small number of studies, and it does not fully cover the extensive literature including field remediation practices, laboratory studies and other areas related to remediation/redevelopment of contaminated areas.
However, we would like to stress that our review is a systematic review. A review is not systematic because of its coverage of a large number of papers or an exhaustive amount of literature (i.e. critical review, overview), but because their rigorous methodological approach providing the highest quality of evidence synthesis for a specific and targeted research questions. A systematic review “seeks to systematically search for, appraise and synthesis research evidence… It is transparent in the reporting of its methods to facilitate others to replicate the process” (Grant and Booth, 2009). The general aim of systematic reviews is to answer questions, which could not be answered by single studies (or other types of reviews), and to define future priorities for research and policy (Page et al., 2021). In comparison to other types of reviews not based on a rigorous methodology, systematic reviews provide a more ‘objective’ assessment of the literature, thanks to a systematic search, objective eligibility criteria and risk of bias assessment. The number of studies identified by the applied methodology is not a useful indicator to assess its applicability, and it is not possible to retrospectively change and modify the approach. If a systematic review yields a small number of studies, it usually indicates the lack of eligible literature, which is an important finding for future priorities. We would also add here that this small number was, to a certain extent expected as we were aware that very few monitoring studies are published in the public domain (see Lines 115 to 124):
“Monitoring reports are often held by consultants and not in the public domain [26]. Indeed, several recent reviews have reported that the lack of full-scale remediation studies in the literature is hampering an evaluation of the outcomes of remediation [19,27–28]. Instead, these reviews focused on laboratory experiments or pilot demonstrations on site, which tend to dominate the literature due to the focus on development of new remediation technologies. Those publications that examine outcomes related to human health tend to cover the pre-remediation risk assessment and/or rely on data modelling exposure pathways or risk (e.g. [2,3,20,23]). However, there is a need to examine the evidence related to full-scale remediation and redevelopment of contaminated sites and their efficacy in terms of reducing harm to human health.”
and 681 to 687:
“Our literature search found many studies reporting on laboratory, modelling or pilot studies rather than full-scale remediation, and the lack of long-term evaluations following full-scale remediation is highlighted as a weakness in the literature in previous reviews focused on the efficacy of remediation for environmental outcomes [19, 27-28]. In some countries, detailed remediation strategies, with well-defined success criteria and long-term monitoring programmes, require regulatory approval prior to the commencement of remediation which are often not easily accessible.”
The review paper raises the important topic of the impacts of contaminated land on human health. It follows the PRISMA methodology and a research process is comprehensively described.
Many thanks for this positive feedback.
Please distinguish more in the introduction what is the aim of the research and what are the research questions.
We have added an aim for the review in to Lines 125 to 127, the research questions are still set out in Lines 132 to 135..
Please elaborate further the research background, present the current state of the research field. I suggest expanding the review with more articles. Maybe change the search criteria - 56 references is not enough for a review article.
We have addressed a similar comment already, this is a systematic review, and as such follows a specific methodology and would repeat our previous response here:
We agree that our review was only able to include a small number of studies, and it does not fully cover the extensive literature including field remediation practices, laboratory studies and other areas related to remediation/redevelopment of contaminated areas.
However, we would like to stress that our review is a systematic review. A review is not systematic because of its coverage of a large number of papers or an exhaustive amount of literature (i.e. critical review, overview), but because their rigorous methodological approach providing the highest quality of evidence synthesis for a specific and targeted research questions. A systematic review “seeks to systematically search for, appraise and synthesis research evidence… It is transparent in the reporting of its methods to facilitate others to replicate the process” (Grant and Booth, 2009). The general aim of systematic reviews is to answer questions, which could not be answered by single studies (or other types of reviews), and to define future priorities for research and policy (Page et al., 2021). In comparison to other types of reviews not based on a rigorous methodology, systematic reviews provide a more ‘objective’ assessment of the literature, thanks to a systematic search, objective eligibility criteria and risk of bias assessment. The number of studies identified by the applied methodology is not a useful indicator to assess its applicability, and it is not possible to retrospectively change and modify the approach. If a systematic review yields a small number of studies, it usually indicates the lack of eligible literature, which is an important finding for future priorities. We would also add here that this small number was, to a certain extent expected as we were aware that very few monitoring studies are published in the public domain (see Lines 115 to 124):
“Monitoring reports are often held by consultants and not in the public domain [26]. Indeed, several recent reviews have reported that the lack of full-scale remediation studies in the literature is hampering an evaluation of the outcomes of remediation [19,27–28]. Instead, these reviews focused on laboratory experiments or pilot demonstrations on site, which tend to dominate the literature due to the focus on development of new remediation technologies. Those publications that examine outcomes related to human health tend to cover the pre-remediation risk assessment and/or rely on data modelling exposure pathways or risk (e.g. [2,3,20,23]). However, there is a need to examine the evidence related to full-scale remediation and redevelopment of contaminated sites and their efficacy in terms of reducing harm to human health.”
and 681 to 687:
“Our literature search found many studies reporting on laboratory, modelling or pilot studies rather than full-scale remediation, and the lack of long-term evaluations following full-scale remediation is highlighted as a weakness in the literature in previous reviews focused on the efficacy of remediation for environmental outcomes [19, 27-28]. In some countries, detailed remediation strategies, with well-defined success criteria and long-term monitoring programmes, require regulatory approval prior to the commencement of remediation which are often not easily accessible.”
Reviewer 4 Report
The paper on “Systematic review of the health and equity impacts of remediation and redevelopment of contaminated sites” examined the extent to which the remediation of contaminated sites reduces health risks to new and existing populations. The review sought to answer the question: to what extent do the remediation and any subsequent redevelopment of contaminated sites reduce environmental and health risks to new and existing populations and ecological systems, and are there any effects on equity in terms of the distribution of risks and outcomes? After reviewing this paper, I have comments and suggestions as follows.
1). You mentioned “Due to time constraints, a review protocol was not developed, and the review was not registered. Can you elaborate on these matters, especially on what you meant by the review was not registered?
2). You wrote in your figured method “full-text articles excluded with reasons (n=38)” but I could not find your explanation on the excluded reasons. In general, how you excluded/reduced articles in each step is not clearly explained. I suggest more elaborating on this aspect. If you can use a table (first column—number of total/remaining articles; second column—number of extracted articles; and third column—reasons of extracting/excluding) to present this aspect would be best.
3). In the Abstract, you stated your review used the standard (quality) databases for searching papers, and in the Methods section you listed the following databases “Ovid (Embase, Medline, Global Health, PsycINFO, Cab Abstracts), Scopus, Open Grey, and ProQuest (theses database, ASSIA)”. Besides Scopus, what criteria that you used to assume those databases are good? At this point, I also have another question: why are your standard databases not included “Web of Science”?
4). I suggest using a table showing those chosen sixteen studies (first column—No. [optional]; second column—authors’ name and published year; and third column—titles of those valid articles).
5). Use ‘Landscape Margin’ for Table 3 (like Table 4).
6). Those Table 3,4&5, is it okay to modify the titles of the last two columns ‘Outcomes’ and ‘Results’? Because we know there are three (order) types of results (outputs > outcomes > impacts). So when you titled those two columns like that were confused.
Author Response
The paper on “Systematic review of the health and equity impacts of remediation and redevelopment of contaminated sites” examined the extent to which the remediation of contaminated sites reduces health risks to new and existing populations. The review sought to answer the question: to what extent do the remediation and any subsequent redevelopment of contaminated sites reduce environmental and health risks to new and existing populations and ecological systems, and are there any effects on equity in terms of the distribution of risks and outcomes? After reviewing this paper, I have comments and suggestions as follows.
1). You mentioned “Due to time constraints, a review protocol was not developed, and the review was not registered. Can you elaborate on these matters, especially on what you meant by the review was not registered?
It is a requirement of PRISMA guidelines that a statement regarding the presence or absence of a review protocol is added to all systematic reviews. Indeed, this was requested by the journal. However, we agree that this probably needs some context so we have replaced the sentence with the following in lines 140 to 142:
“Since the review was carried out in a short space of time, it was not possible to develop and register a review protocol in advance, as suggested by guidelines for systematic reviews [30].”
2). You wrote in your figured method “full-text articles excluded with reasons (n=38)” but I could not find your explanation on the excluded reasons. In general, how you excluded/reduced articles in each step is not clearly explained. I suggest more elaborating on this aspect. If you can use a table (first column—number of total/remaining articles; second column—number of extracted articles; and third column—reasons of extracting/excluding) to present this aspect would be best.
Apologies, this information was presented in the PRISMA Flow Diagram, as per PRISMA guidelines, but it was not clear that the list beneath this statements were the reasons for exclusion. We have now made this clear by bullet pointing the list as shown below (see Figure 1 in the paper):
3). In the Abstract, you stated your review used the standard (quality) databases for searching papers, and in the Methods section you listed the following databases “Ovid (Embase, Medline, Global Health, PsycINFO, Cab Abstracts), Scopus, Open Grey, and ProQuest (theses database, ASSIA)”. Besides Scopus, what criteria that you used to assume those databases are good? At this point, I also have another question: why are your standard databases not included “Web of Science”?
Our choice of databases was guided by an experienced subject librarian at our institution. We subscribe to Scopus but not Web of Science, and this was felt to be appropriate for our review since comparisons of the two conclude that Scopus is more international and interdisciplinary, both of which are important for our review, also that Scopus has better coverage of biomedical research (Tabacaru, 2019). Finally, unlike Web of Science, Scopus includes health sciences (Falagas et al, 2007). We chose to include Open Grey as we expected that some evaluations relevant to our review might not have been published. We have added an explanation to this effect in lines 151 to 155.
Tabacaru, S. (2019). Web of Science versus Scopus: Journal Coverage Overlap Analysis. Texas A&M University Libraries
Falagas, M.E., Pitsouni, E.I., Malietzis, G.A. and Pappas, G. (2008), Comparison of PubMed, Scopus, Web of Science, and Google Scholar: strengths and weaknesses. The FASEB Journal, 22: 338-342. https://doi.org/10.1096/fj.07-9492LSF.
4). I suggest using a table showing those chosen sixteen studies (first column—No. [optional]; second column—authors’ name and published year; and third column—titles of those valid articles).
Thank you for this suggestion, we have added this in as Table 2.
5). Use ‘Landscape Margin’ for Table 3 (like Table 4).
We could not see what this was referring to – it may be that this has already been changed by MDPI.
6). Those Table 3,4&5, is it okay to modify the titles of the last two columns ‘Outcomes’ and ‘Results’? Because we know there are three (order) types of results (outputs > outcomes > impacts). So when you titled those two columns like that were confused.
Thank you for this observation. The column Outcomes referred to how the outcomes were assessed (e.g. lead levels in blood), this has been clarified by changing the column heading in Tables 3 to 5 to ‘Outcome measures’.
Reviewer 5 Report
I have three concerns about the study.
The first is that Web of Science is not one of the databases to be searched. Since Web of Science is one of the most important databases for academic papers, the authors need to explain why they do not include it.
The second problem is that in the introduction part the authors seem to focus on the contaminated land in urban areas, but in the search strategy, the authors also search for ‘agricultural’ and ‘farm’. Maybe the authors should explain this.
Besides the Appendix is very difficult to understand, I suggest the author explain it in the quality appraisal part.
Author Response
The first is that Web of Science is not one of the databases to be searched. Since Web of Science is one of the most important databases for academic papers, the authors need to explain why they do not include it.
Our choice of databases was guided by an experienced subject librarian at our institution. We subscribe to Scopus but not Web of Science, and this was felt to be appropriate for our review since comparisons of the two conclude that Scopus is more international and interdisciplinary, both of which are important for our review, also that Scopus has better coverage of biomedical research (Tabacaru, 2019). Finally, unlike Web of Science, Scopus includes health sciences (Falagas et al, 2007). We chose to include Open Grey as we expected that some evaluations relevant to our review might not have been published. We have added an explanation to this effect in lines 151 to 155.
Tabacaru, S. (2019). Web of Science versus Scopus: Journal Coverage Overlap Analysis. Texas A&M University Libraries
Falagas, M.E., Pitsouni, E.I., Malietzis, G.A. and Pappas, G. (2008), Comparison of PubMed, Scopus, Web of Science, and Google Scholar: strengths and weaknesses. The FASEB Journal, 22: 338-342. https://doi.org/10.1096/fj.07-9492LSF.
The second problem is that in the introduction part the authors seem to focus on the contaminated land in urban areas, but in the search strategy, the authors also search for ‘agricultural’ and ‘farm’. Maybe the authors should explain this.
This is because the contamination might have had an agricultural origin or use (e.g. pesticide manufacture), but an urban impact. For example, the contamination may have resulted from aerial deposition onto an urban area, or the land may have been agricultural in the past but as the urban area has expanded it is now urban.
Besides the Appendix is very difficult to understand, I suggest the author explain it in the quality appraisal part.
We provide a link to the method for the quality appraisal. We feel it is beyond the scope of the paper to provide a detailed explanation of this as it is complex and would be repeating information in the link provided. We have however added a brief summary to lines 205 to 207:
“This tool assesses the quality of studies against a series of criteria including study design, risk of bias, recruitment and withdrawal of participants (a detailed method can be found in the link above).”
Round 2
Reviewer 4 Report
The authors have addressed my concerned comments and carefully revised their manuscript.
This manuscript is a resubmission of an earlier submission. The following is a list of the peer review reports and author responses from that submission.
Round 1
Reviewer 1 Report
The authors give the literature review of contaminated soil and its impact on public health. Although there are no sufficient reports on field remediation, only 16 papers are insufficient for a critical review. I suggest the authors can give more related reports, including laboratory work. In addition, the discussion on future work is not provided in the review. In summary, I don't recommend accepting this review, at least in present status.
Reviewer 2 Report
This article presents a review about on the health-related outcomes in humans following remediation and redevelopment of contaminated sites. So, it can be interesting for international scientific community. However, I see some shortcomings which the authors should consider.
Specific comments
Lines 121-122: Remove the sentence: “Due to time constraints, we did not develop a review protocol or register the review”.
Lines 125: Remove the sentence: “based on [26]” by “based on De Sario [26]”.
Table 1: gives the explanation of the star (*) after the table.
In the Results section, add, if possible, world map with all the selected sites where you can differentiate the chosen ones. This can help for a global view.
Lines 192-200 must go to methods.
Lines 221: “…. Some papers” you should list some of them or the readers can choose wrong one’s for you.
Your tables are not easy to read, please have a regard on the form.
Sometime, the authors argue too much and repeat information, maybe is for a better understanding. However, you must be precise and concise.
Reviewer 3 Report
The authors have done a lot of work to find articles that meet the specified criteria for assessing the impact of contaminated soil on public health. Using the specified algorithm for search, 6903 articles were analyzed and only 54 of them contained the information the authors needed.
The purpose of this article remained unclear to me. Why did the authors analyze the published data on this topic? Did they want to show the absence of clear algorithms for conducting research in the field of assessing the environmental risk of contaminated areas? Do the authors plan to propose a algorithm based on the review? Otherwise, it seems that the topic of manuscript is not of interest to the scientific community.
In addition, there are some comments on the text of the article.
- L.64: Poorly worded sentence. What do authors mean by” inorganic metals”? Among the elements in parentheses (e.g. As, Cd, Pb), only As belongs to metalloids
2.L.207: BLL - occurs for the first time in the text, abbreviation expansion is required
- Table 5: CI - abbreviation expansion and explanation in the text is required.
Reviewer 4 Report
I cannot conclude on the advantage of this work. the manner of treating the data does not give me a clear idea of the usefulness of this bibliography. no statistical study was presented and sometimes the tables were difficult to read.
Please to change these sentences without
…use the expression we,...We searched standard academic d……..
..we did not develop a review protocol or register the review.
…We define a contaminated site as:
… :We note that 104 the later studies at Bunker Hill did assess the independent impact of remediation and 105 public health campaigns
… We undertook, a systematic search strategy, followed by title and abstract screening, full-te
46 : the WHO ??